# Effects of Plyometric Training with Resistance Bands on Neuromuscular Characteristics in Junior Tennis Players

**DOI:** 10.3390/ijerph20021085

**Published:** 2023-01-07

**Authors:** Dario Novak, Iva Loncar, Filip Sinkovic, Petar Barbaros, Luka Milanovic

**Affiliations:** Faculty of Kinesiology, University of Zagreb, 10000 Zagreb, Croatia

**Keywords:** resistance band, neuromuscular training, tennis

## Abstract

The purpose of this study is to investigate the effect of 6 weeks (conducted twice per week for a total of 12 sessions) of plyometric training with resistance bands on different neuromuscular characteristics among the sample of junior tennis players. Thirty junior tennis players between the ages of 12 and 14 years (age 13.5 ± 1.8 years; weight 51.3 ± 12.5 kg; height 162.7 ± 12.6 cm) were allocated to either the control group (standard in-season regimen) (CG; *n* = 15) or the experimental group, which received additional plyometric training with resistance bands (TG; *n* = 15). Pre- and post-tests included: anthropometric measures; 20 m sprint time (with 5, 10, and 20 m splits), squat jump (SQ Jump); vertical countermovement jump (CMJ); vertical countermovement jump with arm swing (CMJ_free arms); single leg (left) countermovement jump (CMJ_L); single leg (right) countermovement jump (CMJ_R); standing long jump (L_Jump); single leg (left) triple jump (SLTH-L); single leg (right) triple jump (SLTH-R); generic change of direction speed (CODS) (20Y test and T-test); reactive agility test (WS-S). After the training intervention, the TG showed significant (“*p* < 0.05”) improvements in CMJ (F = 7.90, *p* = 0.01), CMJ_L (F = 5.30, *p* = 0.03), CMJ_R (F = 11.45, *p* = 0.00), and SLTH-L (F = 4.49, *p* = 0.04) tests. No significant changes were observed in the CG after the training intervention. Our findings provide useful information for coaches to create a wide range of tennis-specific situations to develop a proper performance, especially for their player’s neuromuscular fitness.

## 1. Introduction

Speed-explosive properties are speed, change of direction speed (CODS), and explosive power, and they represent a set of motor abilities very important for success in tennis. These abilities are treated jointly, due to several common characteristics: they use the same energy resources, similarly stimulate the nervous system, have common factors on which the level of a particular ability depends, and meet the same prerequisites for intensive training of a particular motor ability [1]. In addition, it is considered that athletes with more pronounced speed-explosive properties find it easier to control their body in urgent training and competition situations, which greatly contributes to the game, but also to the prevention of injuries [2].

Athletes as well as their coaches are trying to find new ways to improve certain motor abilities and, thus, improve results in certain sports. This includes the method of plyometrics as one of the most effective methods for the development of different types of explosive power and can be explained as any type of training in which eccentric–concentric muscle work occurs [3]. Many studies agree that plyometrics involves specific exercises that cause significant stretching of a muscle that is under eccentric contraction and is followed by strong concentric contraction [4,5,6]. Such a mechanism serves to develop a strong movement in a short period of time. In addition, a very significant element of the plyometric system is the reactive ability of the apparatus to move. This means the summary contribution of the muscle stretching reflex, with the muscle contracting strongly immediately after stretching [7]. Plyometrics increase neuromuscular coordination by training the nervous system and making movements more automatic during activity (training effect). This is known as reinforcing a motor pattern and creating automation of activity, which improves neural efficiency and increases neuromuscular performance [7]. Due to all the above, the influence of plyometric training on biomechanical and physiological parameters in tennis is increasingly being researched. 

A review of the literature shows that the use of plyometric training in everyday tennis training significantly affects the ability to change direction; more precisely, better results are achieved in change of direction tests (T-test, 505 test) [5] and in the sidestep test [8]. The application of combined plyometric and tennis training also showed an improvement in speed, more precisely, a 20 m sprint [5] and 12 m sprint [6]. In addition to the ability to change direction and speed, combined plyometric and tennis training affect jumping where significantly better results have been found in broad jump and countermovement jump [5]. In addition, using a combination of plyometric and tennis training, an improvement in tennis player strength was found in the upper extremities [5] as well as in the lower extremities and service speed, which is a very important component of tennis success [6]. Using additional plyometric training 2–3 times a week with daily tennis training can significantly affect jumping where it is seen how to achieve better results in long jump and triple jump [9] as well as in vertical jumps [10]. However, no studies have simultaneously examined the contribution of the plyometric training with bands on different neuromuscular factors. The elastic band presents a new material for plyometric training with load-like weight machines, but it is less expensive and is simple to implement. This kind of training can activate all relevant muscles and requires little time [11]. There is a need for this type of scientific research. 

Accordingly, in the present study, we investigated the effects of plyometric training with resistance WearBands™ bands on neuromuscular characteristics among a sample of junior tennis players.

## 2. Materials and Methods

### 2.1. Participants

Thirty junior tennis players (15 boys and 15 girls) between the ages of 12 and 14 years (age 13.5 ± 1.8 years; weight 51.3 ± 12.5 kg; height 162.7 ± 12.6 cm) were randomly allocated to either the control group (standard in-season regimen) (CG; *n* = 15; 8 boys and 7 girls) or the experimental group, which received an additional plyometric training with resistance bands (TG; *n* = 15; 7 boys and 8 girls). To participate in the study, all subjects had to meet the criteria that they are healthy, physically active players who train at least three times a week and compete in regional, national, or international tournaments. All the participants have from 6 to 8 h of tennis training per week, and during the training program, strength training was prohibited. All participants were informed about the subject and goal of the study, and the subjects and their parents gave written consent to participate. The complete testing protocol was explained to them in detail with special emphasis on the fact that the study requires certain additional effort and presents a risk of injury that is the same as during the standard training process or competition. The research was conducted in accordance with the Declaration of Helsinki and approved by the Ethics Committee of the Faculty of Kinesiology of the University of Zagreb (protocol code 34; date of approval 10 May 2022). 

### 2.2. Measurements

A week before starting with the training program, initial tests were provided on each subject on the same day. Each subject was tested in the same order and recorded with the same equipment by the same investigators. There were three investigators and each of them was charged for the specific group of tests. For basic anthropological variables, a bioelectrical impedance analyzer was used (HBF-500, Kyoto, Japan). The time completing the 20 m straight line dash with 5 m, 10 m, and 20 m sprint times as well as the generic CODS (20Y test and T-test) were measured with the Powertimer Newtest system (Oulu, Finland). The reactive agility test in the sagittal plane was tested with the Wireless Training Timer SEM Witty (Microgate, Bolzano, Italy). For measuring standing long jump (L_Jump), single leg (left) triple jump (SLTH-L), and single leg (right) triple jump (SLTH-R), we used a tape measure, and for the squat jump (SQ Jump), vertical countermovement jump (CMJ), vertical countermovement jump with arm swing (CMJ_free arms), single leg (left) countermovement jump (CMJ_L), and single leg (right) countermovement jump (CMJ_R), we used the Microgate Optogait system (Microgate, Bolzano, Italy).

### 2.3. Study Design and Procedure

Physical tests were carried out before (week 0) and after the training period (end of the sixth week), including anthropometric measures, 20 m sprint time (with 5, 10, and 20 m splits), squat jump (SQ Jump), vertical countermovement jump (CMJ), vertical countermovement jump with arm swing (CMJ_free arms), single leg (left) countermovement jump (CMJ_L), single leg (right) countermovement jump (CMJ_R), standing long jump (L_Jump), single leg (left) triple jump (SLTH-L), single leg (right) triple jump (SLTH-R), generic CODS (20Y test and T-test), and reactive agility tests (WS-S) [12]. Mind Mapping for the research procedure is shown in Figure 1. 

As the subjects were young athletes, the training program needed to be adapted to their age and abilities. In addition to standard technical-tactical training, the control group performed a plyometric training program without WearBands, while the experimental group, in addition to standard technical-tactical training, performed the same plyometric training program but with WearBands (Table 1). With such a combination, the effect obtained was that both groups were equal in terms of the total load volume, and they had the same number of training units and training hours.

The 6-week plyometric training is composed of low (i.e., ankle cone hops) and moderate levels of plyometric exercises (i.e., tuck jumps) (Table 1). It includes different kinds of vertical, horizontal, and lateral jumps and hops that are scheduled in each training. As the fact that elastic bands are used, only lower-body exercises are included in the program. Table 1 shows the training schedule that is described by the number of weeks, names of the exercises, numbers of sets and repetitions, and the rest period. Subjects mostly performed 3 to 4 sets of 4 to 6 exercises with 5 to 10 repetitions with maximum intensity. Participants were instructed to perform all exercises with maximal effort. Depending on the exercise, the rest period was between 15 and 60 s between sets and 60 to 120 s between the exercises. The duration of the training was between 30 and 45 min including the warm-up period and was led by a certified strength and conditioning coach. The warm-up protocol included light-intensity running over 10 lengths of 20 m, after which followed dynamic stretching exercises for a total of 15 min (lateral movements, skipping, jumping, lunges, and, finally, 4 lengths of sub-maximum acceleration).

### 2.4. Experimental Protocol

The multi-patented WearBands™ Dynamic Gravitational Resistance Training System applies gravitational, multi-planar, and multi-directional resistance during sport-specific movement at or near full-speed. By applying multi-planar resistance, the system allows the athlete to maintain their normal center of gravity while amplifying neuromuscular stimulation during sport-specific movement. Unlike more traditional “bungee” resistance systems, which apply mostly single-plane, single-direction sheer resistance, WearBands™ improves the force production into and through the ground during any movement in any direction. This unique ability allows sport-specific change of direction training (in any direction), as well as aiding first-step quickness, acceleration, and speed development. The system’s neuromuscular stimulation and feedback also aids an athlete’s reactive ability. By allowing an athlete to move in any direction at any speed with little or no restrictions, while simultaneously amplifying neuromuscular stimulation, the athlete can train precise sports-specific movement in a way not possible before. The elastic bands were of differing elasticity. Green elastic bands (easy resistance), gray elastic bands (low resistance), and yellow elastic bands (moderate resistance) were used for the first and second weeks, the third and fourth weeks, and the fifth and sixth weeks, respectively, to ensure progression. For each color, the training sessions were performed by stretching the band to 75% the first week and 100% for the second week. The elastic band plyometric training began with a resistance of 3.0 kg, with an increase of 1 kg every two weeks to reach a final resistance of 5.0 kg, and the resistance was derived from the manufacturer’s manual, based on the elongation of the band.

### 2.5. Statistical Analysis

Basic descriptive parameters (mean—$\overset{—}{x}$ standard deviation—SD) were used to describe variables for each group and measurement. A 2 × 2 (time*group) mixed-model ANOVA was used to assess the influence of the training program on CG and TG. The partial ŋ2 coefficient was used as an indicator of effect size. Tukey’s post hoc test was performed for further analysis of variables with significant interactions. Statistical analysis was performed with the use of Statistica 14.0.1.25 (TIBCO software, Inc., Sydney, Australia). The level of statistical significance was set at “*p* < 0.05”.

## 3. Results

Table 2 points out descriptive statistics parameters. In the CG, there was an improvement in results after the training program; only CMJ and SLTH-R values were lower at the final testing in comparison to initial measurement. TG results showed an improvement in all observed test after finishing the training program. Statistically significant interactions were determined in CMJ (F = 7.90, *p* = 0.01), CMJ_L (F = 5.30, *p* = 0.03), CMJ_R (F = 11.45, *p* = 0.00), and SLTH-L (F = 4.49, *p* = 0.04) tests. Tukey’s post hoc test was used to further analyze significant interactions between variables (Table 3). In tests CMJ, CMJ_L, CMJ_R, and SLTH-L, interactions in TG indicate significant differences after the plyometric training program with resistance bands. Post hoc results of CG did not show significant differences in the observed tests between initial and final measurement.

## 4. Discussion

This study aimed to investigate the effect of the plyometric training with resistance bands on different neuromuscular characteristics among the sample of junior tennis players. We can conclude that the plyometric training with resistance bands significantly increased certain lower-body vertical and horizontal jumps over the regular plyometric training alone. 

### 4.1. The First-Step Quickness, Acceleration, and Speed

In the sport of tennis, players must be able to react as fast as possible to actions performed by the opponent, where reaction time, initial acceleration, and ability to change direction play an important role [5]. The game is characterized by high-intensity efforts in terms of the first-step quickness, acceleration, and speed. The first-step quickness is one of the most important factors for the player to reach an effective hitting position [12]. However, the ability to accelerate within a short distance is an essential requirement for handling the ball correctly to successfully solve the game situations [12,13]. Initial acceleration can be referred to as the first 10 m and 20 m of a sprint [5]. The results of our study show that the experimental program did not affect any of the speed components in the horizontal direction among the sample of young tennis players. Several studies conducted on tennis players showed improvements in the first-step quickness, acceleration, and speed performance after plyometric training [5,6,10]. Therefore, regarding the non-significant improvement in the sprint of 5, 10, and 20 m as a result of applied plyometric training with resistance bands training, and the fact that the first-step quickness, acceleration, and speed are highly important for the success in the tennis match [12], we may hypothesize that in future studies, improvements may occur by more emphasis on horizontal power within the training program. It is possible that the contents of the plyometric program of this study did not significantly emphasize the maximum horizontal component of the performance.

### 4.2. Lower-Body Explosive Power

The results of the research indicate that there were significant changes in the results of tests for estimating the explosive power of the lower extremities in the horizontal and vertical components of jumping. It can be concluded that plyometric training with elastic resistance does affect the improvement of performance in explosively strong properties of horizontal and vertical type. It is to be assumed that plyometric training with resistance bands emphasizes eccentric stimuli with an emphasis on performance speed and force in the performance of tasks, especially in the sample of young tennis players. This is one of the first studies to conduct plyometric training with resistance bands; therefore, comparisons are difficult as previous studies were focused exclusively on plyometric training conducted mostly with mature players [5,6]. For example, a previous study reported a significant improvement in drop jump (DJ; 15%) and lower-extremity maximum isometric force (11%) after 8 weeks of training [6]. One of the few studies of the impact of plyometric training for physical abilities in young tennis players showed how plyometric training seems to be an appropriate stimulus for improving physical qualities in tennis players. It demonstrates the importance of specific power training for enhancing the explosive actions of tennis players [5]. Similar findings were established that plyometric training could increase horizontal jumping performance by 1.4% to 7%, with less improvement than vertical jumping, however [14]. In addition, our results might be explained by having a combination of lateral, horizontal, and vertical direction drills, in contrast to previous studies, in which the number of vertical direction drills was higher [15].

### 4.3. Change of Direction Speed and Reactive Agility 

In our study, significant improvements were not recorded in generic CODS performance (measured by T-test and 20-yard ability to change direction test) after plyometric training with resistance bands. Tennis is an extremely dynamic sport in which players perform 300–500 high-intensity efforts during a best-of-three-sets match [16]. Therefore, CODS is considered as one of the key performances in tennis. CODS comprises the acceleration phase, deceleration phase, change of direction, and reacceleration in the other direction [17]. Improvements in CODS performance after plyometric training are supported by several studies conducted on tennis players [5,10,18]. In other studies, young tennis players also showed significant improvements in the tests of generic CODS after multiple weeks of neuromuscular training [19,20,21,22]. On the contrary, our plyometric training with resistance bands did not cause an improvement in generic CODS performance. It is possible that the contents of the plyometric program of this study did not significantly emphasize the maximum horizontal component of the performance and, therefore, significant improvements were not recorded in generic CODS performance after plyometric training with resistance bands. 

In our study, significant improvements were not recorded in sport-specific reactive agility performance after plyometric training with resistance bands. A limited number of studies evaluated the effects of training on sport-specific reactive agility performance, measured by sport-specific tests [6]. For example, one study investigated the effect of speed, ability to change direction, and quickness training on reactive agility in soccer players [2]. Players significantly improved the ability to change direction performance by 4.2%, but the authors suggest that improvement occurred due to improvements recorded in the speed over 5 m and not due to faster decision-making ability [2]. There are few studies that use tests to assess the reactive component of ability to change direction. In one such study, the authors concluded that plyometric training improved fitness characteristics that rely more on reactive strength and the powerful push-off of legs such as lateral reaction time, 4 m lateral and forward sprints, drop jump, and maximal force [6].

### 4.4. Limitations

This study had a number of limitations, which are discussed below. First, the subjects involved in this study were selected youth tennis players in a very sensitive and crucial developmental phase. Secondly, we did not evaluate the biological age of the participants, which is known to influence neuromuscular performance. Thirdly, we did not have the possibility to look at the mental and physical fatigue that may have occurred during the testing process; therefore, it could have potentially affected the most effective movement execution.

## 5. Conclusions

In summary, the present study suggests that there is a positive effect of the plyometric training with resistance bands mostly on lower-body vertical and horizontal jumps, but not on the first-step quickness, acceleration, speed, change of direction speed, and reactive agility characteristics in junior tennis players’ players. Our findings provide useful information for coaches to create a wide range of tennis-specific situations to develop a proper performance, especially for their player’s neuromuscular fitness. Additional studies are needed to identify interventions that can increase sport-specific neuromuscular fitness with the ultimate goal of achieving better performance.

## Figures and Tables

**Figure 1 ijerph-20-01085-f001:**
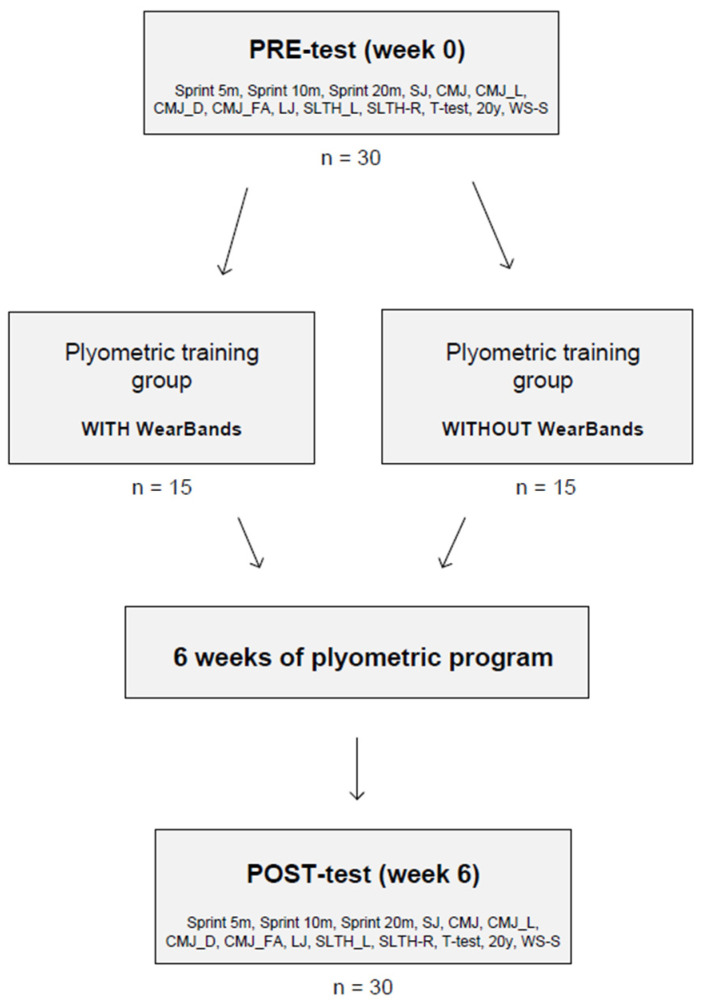
Mind Mapping for the research procedure.

**Table 1 ijerph-20-01085-t001:** Six-week plyometric training program.

Training Week	Exercise	Sets × Reps	Rest (s)
1	Ankle cone hops	3 × 10	15–30/90
	Ankle cone hops side to side	3 × 10	15–30/90
	CMJ	4 × 5	15–30/90
	Broad jumps	4 × 5	15–30/90
2	1-leg ankle hops forward	3 × 10	30–60/90–120
	CMJ	3 × 8	30–60/90–120
	Continuous broad jumps	3 × 2 × 3	30–60/90–120
	Lateral bounds + stick	3 × 6	30–60/90–120
	2–1 Hurdle hops forward (20–30 cm)	3 × 10	30–60/90–120
3	1-leg ankle hops lateral	3 × 10	30–60/90–120
	CMJ	3 × 10	30–60/90–120
	1:2 broad jumps	3 × 4 e.l.	30–60/90–120
	Zig zag bounds + stick	3 × 8	30–60/90–120
	2–1 Hurdle hops lateral (20–30 cm)	3 × 10	30–60/90–120
4	1-leg square ankle hops 1-leg	3 × 8 e.l.	30–60/90–120
	CMJ	3 × 5 e.l.	30–60/90–120
	Continuous broad jumps	3 × 3 × 3	30–60/90–120
	Lateral bounds (1–1-stick)	3 × 8 e.l.	30–60/90–120
	2–1 Multidirectional hurdle	3 × 10	30–60/90–120
	hops Tuck jumps	3 × 8	30–60/90–120
5	1-leg square ankle hops 1-leg	3 × 12 e.l.	30–60/90–120
	CMJ	3 × 6 e.l.	30–60/90–120
	1:2 Broad jumps	3 × 5 e.l.	30–60/90–120
	Zig zag bounds (1–1-stick)	3 × 8 e.l.	30–60/90–120
	2–1 Multidirectional hurdle hopes	3 × 10	30–60/90–120
	Tuck jumps	3 × 10	30–60/90–120
6	Ankle cone hops	3 × 10	15–30/90
	Ankle cone hops side to side	3 × 10	15–30/90
	CMJ	4 × 5	15–30/90
	Broad jumps	4 × 5	15–30/90

**Table 2 ijerph-20-01085-t002:** Descriptive statistics for both groups in two time points of measurements and results of 2 × 2 mixed-model ANOVA for each variable.

Variable	Control Group		Experimental Group		Interaction TIME × GROUP		
	Initial Testing	Final Testing	Initial Testing	Final Testing			
	$\overset{—}{x}\;\pm \;\mathrm{SD}$	$\overset{—}{x}\;\pm \;\mathrm{SD}$	$\overset{—}{x}\;\pm \;\mathrm{SD}$	$\overset{—}{x}\;\pm \;\mathrm{SD}$	F	P	Partial ŋ2
Sprint 5 m	1.83 ± 0.14	1.75 ± 0.15	1.82 ± 0.11	1.71 ± 0.10	0.45	0.51	0.05
Sprint 10 m	2.86 ± 0.24	2.65 ± 0.24	2.83 ± 0.19	2.58 ± 0.13	0.62	0.44	0.04
Sprint 20 m	4.46 ± 0.46	4.29 ± 0.46	4.45 ± 0.34	4.19 ± 0.24	0.84	0.37	0.07
SQ_Jump	27.38 ± 7.81	27.63 ± 6.49	24.74 ± 5.60	26.8 ± 4.85	2.02	0.17	0.11
CMJ	27.81 ± 7.24	27.36 ± 7.35	24.72 ± 5.79	27.08 ± 4.74	7.90	0.01 *	0.32
CMJ_L	13.01 ± 3.82	13.55 ± 3.61	11.46 ± 3.26	13.98 ± 3.56	5.30	0.03 *	0.25
CMJ_R	13.41 ± 3.93	13.61 ± 3.35	12.15 ± 3.26	14.79 ± 3.02	11.45	0.00 *	0.42
CMJ_Free arms	30.47 ± 7.50	30.79 ± 6.81	27.00 ± 6.47	29.8 ± 6.02	3.38	0.08	0.17
L_Jump	177.99 ± 26.69	181.76 ± 27.83	180.17 ± 18.40	185.1 ± 20.23	0.05	0.82	0.00
SLTH-L	476.97 ± 117.11	484.9 ± 113.30	471.82 ± 71.25	503.00 ± 62.25	4.49	0.04 *	0.20
SLTH-R	484.03 ± 110.98	482.93 ± 123.24	477.6 ± 83.18	512.1 ± 68.57	3.11	0.10	0.15
T-test	12.48 ± 1.39	12.44 ± 2.00	12.3 ± 1.08	11.83 ± 0.87	2.14	0.16	0.11
20 yards	5.6 ± 0.49	5.5 ± 0.52	5.65 ± 0.35	5.5 ± 0.39	0.51	0.49	0.03
WS-S	17.72 ± 2.98	16.68 ± 2.59	18.44 ± 2.42	16.92 ± 2.25	0.20	0.66	0.02

Legend: Sprint 5 m—result of split time on 5 m; Sprint 10 m—result of split time on 10 m; Sprint 20 m—result of 20 m speed and acceleration test; SQ_Jump—squat jump; CMJ—countermovement jump with arms set on hips; CMJ_L—single leg (left) countermovement jump with arms set on hips; CMJ_R—single leg (right) countermovement jump with arms set on hips; CMJ_free arms—countermovement jump with free arms swing; L_Jump—long jump, SLTH-L—single leg (left) triple hop; SLTH-R—single leg (right) triple hop; T- test—ability to change direction test; 20 yards—turn ability to change direction test; WS-S—Witty Sem sagittal plane; $\overset{—}{x}$—arithmetic mean; SD—standard deviation; F—F value; *p*—significance indicator; Partial ŋ2—measure of effect size; *—significant interaction (*p* < 0.05).

**Table 3 ijerph-20-01085-t003:** Tukey’s post hoc results for variables with significant interactions.

CMJ							CMJ_L						
Interaction	Group	Time	1	2	3	4	Interaction	Group	Time	1	2	3	4
1	1	1		0.90	0.86	1.00	1	1	1		0.78	0.88	0.87
2	1	2	0.90		0.93	1.00	2	1	2	0.78		0.70	0.97
3	2	1	0.86	0.93		0.02 *	3	2	1	0.88	0.70		0.00 *
4	2	2	1.00	1.00	0.02 *		4	2	2	0.87	0.97	0.00 *	
**CMJ_R**							**SLTH L**						
Interaction	Group	Time	1	2	3	4	Interaction	Group	Time	1	2	3	4
1	1	1		0.98	0.94	0.69	1	1	1		0.74	1.00	0.86
2	1	2	0.98		0.89	0.77	2	1	2	0.74		1.00	0.93
3	2	1	0.94	0.89		0.00 *	3	2	1	1.00	1.00		0.01 *
4	2	2	0.69	0.77	0.00 *		4	2	2	0.86	0.93	0.01 *	

Legend: CMJ—countermovement jump with arms set on hips; CMJ_L—single leg (left) countermovement jump with arms set on hips; CMJ_R—single leg (right) countermovement jump with arms set on hips; SLTH-L—single leg (left) triple hop; *—significant interaction (*p* < 0.05).

## Data Availability

Data available on request.

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
