# Peer review of "Effects of Plyometric Training with Resistance Bands on Neuromuscular Characteristics in Junior Tennis Players"

_ijerph, 2023, doi:10.3390/ijerph20021085_

Round 1

Reviewer 1 Report

Thanks for the opportunity to review this manuscript. I believe the authors are focused on high-level athletic training methods. This study discusses the training effect of Plyometric Training combined with Resistance Bands. I have some comments that are listed below.

1. Introduction

“Speed-explosive properties are speed, agility and explosive power, and they represent a set of motor skills very important for success in tennis. These abilities are treated jointly………”. "Speed, agility and explosive power" are motor abilities or motor skills?

More research on WearBandsTH as auxiliary training equipment should be added. And the importance of WearBandsTH as an auxiliary tool in training should be emphasized.

2. We consider the "T test", "505 test" and "Y test" to be tests that evaluate the ability to change direction quickly. The manuscript wants to express whether the agile ability is significantly improved or the ability to change direction quickly is improved. We think this is a different concept. It is suggested to modify part of agility to the ability to change direction.

3. The description of the participants is unclear. What are the criteria for a participant "Top Junior Tennis Player"? What was the ratio of boys to girls among the participants?

4. The experimental plan is not described in detail. Did the CG group perform plyometric training without WearBandsTH? Whether the CG group did not perform plyometric training, a plyometric training group without WearBandsTH should be added. If the CG group is a plyometric training group without WearBandsTH, it is necessary to consider whether there is an interaction between WearBandsTH and plyometric training.

5. The description in the "Methods" section is insufficient, and the difference between the experimental protocols of the TG and the CG is not clear.

It seems that the measured variables in Table 5 are different from those described in the Methods section.

Providing a mind Mapping for the research procedure picture would be better.

How is the resistance level of the resistance band controlled in the study design?

The introduction of WearBandsTH can be placed in "Introduction", or a new title "Experimental Protocol" can be set up, and the introduction of WearBandsTH can be placed in the "Experimental Protocol" section.

6. The "Measurement" section should let readers know the basic norms and standards of measurement.

7. In the "Statistical Analysis", have you considered the existence of interaction?

8. The "Table 2" section, is not meant to simply enumerate the basic characteristics of the participants. It is suggested that the variables of the basic characteristics of the players in the two groups can be tested for homogeneity.

9. There are 4 places in the discussion section that use the APA references format, and the format of the references should be unified.

10. Limitations are mentioned in the article, and the "Limitations" section can be listed separately.

11. The beginning and conclusion of the article have "explosive power", but the whole piece is not about explosive power. Agility is also mentioned many times in the article, but the conclusion of the research mainly revolves around the ability to change direction quickly, the practical improvement of acceleration ability and speed ability.

Author Response

Dear Editor and dear Reviewer:

Firstly, since we are resubmitting revised version of our Manuscript on the Christmas day, we are using this opportunity to wish you a very Merry Christmas! Warmest wishes to you and yours!

Secondly, we are pleased to resubmit for publication the revised version of Manuscript entitled "Effects of Plyometric Training with Resistance Bands on Neuromuscular Characteristics in Junior Tennis Players". We appreciate the constructive criticisms of the Editor and the reviewers. We have addressed each of their concerns as outlined below.

Responses to the comments from Reviewer 1:

Thanks for the opportunity to review this manuscript. I believe the authors are focused on high-level athletic training methods. This study discusses the training effect of Plyometric Training combined with Resistance Bands. I have some comments that are listed below.

Thank you these comments. We really appreciate your constructive criticisms.

  1. Introduction

“Speed-explosive properties are speed, agility, and explosive power, and they represent a set of motor skills very important for success in tennis. These abilities are treated jointly………”. "Speed, agility and explosive power" are motor abilities or motor skills?

Absolutely agree with this comment. It should be listed as motor abilities. Done.

More research on WearBands as auxiliary training equipment should be added. And the importance of WearBands as an auxiliary tool in training should be emphasized.

Thank you for these comments. This research is the first research using the WearBands as auxiliary training equipment. However, the importance of WearBands as an auxiliary tool in training should be emphasized.

  1. We consider the "T test", "505 test" and "Y test" to be tests that evaluate the ability to change direction quickly. The manuscript wants to express whether the agile ability is significantly improved or the ability to change direction quickly is improved. We think this is a different concept. It is suggested to modify part of agility to the ability to change direction.

Done.

  1. The description of the participants is unclear. What are the criteria for a participant "Top Junior Tennis Player"? What was the ratio of boys to girls among the participants? 

Done.

  1. The experimental plan is not described in detail. Did the CG group perform plyometric training without WearBands? Whether the CG group did not perform plyometric training, a plyometric training group without WearBands should be added. If the CG group is a plyometric training group without WearBands, it is necessary to consider whether there is an interaction between WearBands and plyometric training.

Thank you for the suggestion regarding the new approach in reporting results of our study. We changed statistical analysis and conducted 2x2 mixed model ANOVA to get a clear insight into interactions of between (control and training group) and within (initial and final testing) factors for each tested variable. Also, we conducted Tukey post-hoc test for variables with determined significant interactions. We agree that this new approach is more suitable for our study design and helps in better understanding of experimental procedures, i.e., the idea and results of our intervention.

  1. The description in the "Methods" section is insufficient, and the difference between the experimental protocols of the TG and the CG is not clear.

Agree. This section has been rewritten and we strongly believe that the difference between the experimental protocols of the TG and the CG is much clearer now.

It seems that the measured variables in Table 5 are different from those described in the Methods section.

Done.

Providing a mind Mapping for the research procedure picture would be better. 

Done.

How is the resistance level of the resistance band controlled in the study design? 

Done.

The introduction of WearBands can be placed in "Introduction", or a new title "Experimental Protocol" can be set up, and the introduction of WearBands can be placed in the "Experimental Protocol" section.

Done.

  1. The "Measurement" section should let readers know the basic norms and standards of measurement.

Done.

  1. In the "Statistical Analysis", have you considered the existence of interaction?

As we already mentioned in response 4, we agree with your suggestions and decided to change statistical analysis to determine potential interactions of between and within factors (2x2 mixed model ANOVA and Tukey post-hoc test for variables with determined significant interactions). Accordingly, we provided new tables (Table 3 and Table 4) in Results section of the manuscript. The most important results are then further analyzed and explained in the accompanying texts under the tables.

  1. The "Table 2" section, is not meant to simply enumerate the basic characteristics of the participants. It is suggested that the variables of the basic characteristics of the players in the two groups can be tested for homogeneity.

Done.

  1. There are 4 places in the discussion section that use the APA references format, and the format of the references should be unified.

Thank you for these comments! The format of the references is unified now.

  1. Limitations are mentioned in the article, and the "Limitations" section can be listed separately.

Done.

  1. The beginning and conclusion of the article have "explosive power", but the whole piece is not about explosive power. Agility is also mentioned many times in the article, but the conclusion of the research mainly revolves around the ability to change direction quickly, the practical improvement of acceleration ability and speed ability.

Absolutely agree with this comment! This section has been re-written, and we believe it is much clearer now.

We thank the editor and the reviewers again for their helpful comments, which we feel have improved our manuscript. We hope that with these modifications, our paper can now be accepted for publication.

Sincerely,

Dario Novak

Reviewer 2 Report

The introduction has to be improved with more investigation up to 2016 - authors have to separate the introduction into paragraphs and need to show the possible physiological impact of plyometric exercises.

In participants - authors have to insert eligibility inclusion and exclusion criteria.

In measurements - for each variable, give sources of data and details of assessment methods.

In procedures - it is necessary to insert more details - warm-up/ authors have to separate the study design and procedures and all the text into paragraphs.

Result - the authors need to improve the formatting of table 3. About the statistics, it is necessary to insert the value of the t-test and the confidence interval of the variables.

Author Response

Dear Editor and dear Reviewer:

Firstly, since we are resubmitting revised version of our Manuscript on the Christmas day, we are using this opportunity to wish you a very Merry Christmas! Warmest wishes to you and yours!

Secondly, we are pleased to resubmit for publication the revised version of Manuscript entitled "Effects of Plyometric Training with Resistance Bands on Neuromuscular Characteristics in Junior Tennis Players". We appreciate the constructive criticisms of the Editor and the reviewers. We have addressed each of their concerns as outlined below.

Responses to the comments from Reviewer 2:

The introduction has to be improved with more investigation up to 2016 - authors have to separate the introduction into paragraphs and need to show the possible physiological impact of plyometric exercises.

Thank you these comments. We really appreciate your constructive criticisms. More recent literature has been added and we’ve separated the introduction into paragraphs and need to show the possible physiological impact of plyometric exercises.

In participants - authors have to insert eligibility inclusion and exclusion criteria.

Done.

In measurements - for each variable, give sources of data and details of assessment methods.

Done.

In procedures - it is necessary to insert more details - warm-up/ authors have to separate the study design and procedures and all the text into paragraphs.

Done.

Result - the authors need to improve the formatting of table 3. About the statistics, it is necessary to insert the value of the t-test and the confidence interval of the variables.

Thank you for these comments. We really appreciate your constructive criticisms. Due to the comments of other reviewers, we decided to change the overall statistical analysis. We conducted 2x2 mixed model ANOVA to get a clear insight into interactions of between (control and training group) and within (initial and final testing) factors for each tested variable. Also, we conducted Tukey post-hoc test for variables with determined significant interactions. This new approach, suggested from other reviewers, is more suitable for our study design and helps in better understanding of experimental procedures, i.e., the idea and results of our intervention. You can find incorporated new tables (Table 3 and 4) in Results section of our manuscript. The most important results are then further analyzed and explained in the accompanying texts under the tables.

We thank the editor and the reviewers again for their helpful comments, which we feel have improved our manuscript. We hope that with these modifications, our paper can now be accepted for publication.

Sincerely,

Dario Novak

Reviewer 3 Report

Effects of Plyometric Training with Resistance Bands on Neuromuscular Characteristics in Junior Tennis Players

First of all, the reviewer would like to thank the authors for their work and efforts in trying to improve sports science knowledge.

General comments to the authors

Overall, this is a nice study that could have great practical application. The authors are commended on their efforts thus far. The study is well designed and well-written, with a great original article evaluating the usefulness of the topic. However, I suggest only small corrections for manuscript.

Abstract

This section is well designed and well-written.

Introduction section

This section is short but it is enough.

Methods section

The authors should add number of the Ethics files

Statistical Analysis is old fashion. The authors shold be made mixed anova to determine within and between groups differences

What about maturation?

Results section

This section is is old fashion

Discussion section

Overall the discussion is well-written and incorporates relevant literature.

Author Response

Dear Editor and dear Reviewer:

Firstly, since we are resubmitting revised version of our Manuscript on the Christmas day, we are using this opportunity to wish you a very Merry Christmas! Warmest wishes to you and yours!

Secondly, we are pleased to resubmit for publication the revised version of Manuscript entitled "Effects of Plyometric Training with Resistance Bands on Neuromuscular Characteristics in Junior Tennis Players". We appreciate the constructive criticisms of the Editor and the reviewers. We have addressed each of their concerns as outlined below.

Responses to the comments from Reviewer 3:

First of all, the reviewer would like to thank the authors for their work and efforts in trying to improve sports science knowledge. 

Thank you these comments. We really appreciate your constructive criticisms.

General comments to the authors

Overall, this is a nice study that could have great practical application. The authors are commended on their efforts thus far. The study is well designed and well-written, with a great original article evaluating the usefulness of the topic. However, I suggest only small corrections for manuscript.

Thank you these comments. We really appreciate your constructive criticisms.

Abstract

This section is well designed and well-written.

Thank you!

Introduction section

This section is short, but it is enough. 

Thank you!

Methods section

The authors should add number of the Ethics files.

There is a protocol code for ethical approval, and it has been added within the Methods section. Thank you for this comment!

Statistical Analysis is old fashion. The authors should be made mixed anova to determine within and between groups differences.

Thank you for your observations, we agree with your statements and suggestions. Due to your comments and comments of other reviewers, we changed overall statistical analysis and conducted 2x2 mixed model ANOVA to get a clear insight into interactions of between (control and training group) and within (initial and final testing) factors for each tested variable. We agree that this new approach is more suitable for our study design and helps in better understanding of experimental procedures, i.e., the idea and results of our intervention. You can find incorporated new tables (Table 3 and 4) in Results section of our manuscript. The most important results are then further analyzed and explained in the accompanying texts under the tables.

What about maturation?

Thank you this comment. We really appreciate your constructive criticisms. Thank you for pointing this out. We definitely agree with your comment. These information have been added within the Limitation section.

Results section

This section is old fashion.

Thank you this comment. We’ve re-written this section according the new statistical analyses..

Discussion section

Overall, the discussion is well-written and incorporates relevant literature.

Thank you these comments. We really appreciate your constructive criticisms.

We thank the editor and the reviewers again for their helpful comments, which we feel have improved our manuscript. We hope that with these modifications, our paper can now be accepted for publication.

Sincerely,

Dario Novak

Round 2

Reviewer 1 Report

Abstract

CODS should be spelled out in full on the first occurrence.

"p < .05" requires uniform formatting.

Introduction

Add research on the use of WearBands™ for muscle training and increase the need for this research.

Participants

How were the 30 subjects divided into two groups?

Have you considered the ratio of male to female in the two groups?

Measurements

After the intervention experiment, when was the measurement data obtained after the intervention training phase?

“There were three of investigators and each of them was charged for the specific group of tests.” Does it mean that the test items are divided into three project groups, and three investigators are responsible for each?

Study Design and Procedure

“The 6-week plyometric training is composed of low and moderate levels of plyometric exercises” What are the criteria for distinguishing between low and moderate levels? The training program needs to reflect the low and moderate levels

“Subjects mostly performed 3 to 4 sets of 4 to 6 exercises with 5 to 10 repetitions with maximum intensity.” How to monitor the implementation of maximum intensity?

Experimental protocol

The study protocol did not address the use of 3 levels of resistance bands during the intervention. This should be supplemented in the study design. Need to explain the resistance parameters of the following three strength elastic bands?

Statistical Analysis

 “p ≤ 0.05 Please confirm ≤ or <.

Results

Table 2. What is the significance of the description of the basic characteristics of the participants? If there are differences in the basic characteristics of the samples, there is no need to continue the analysis.

Table 3. The results section in the manuscript only describes the changes within the group before and after the experiment within the group. Why is there no description of whether there is significance between the groups after the experiment?

Discussion

Many studies mentioned in the introduction have proved that plyometric training is beneficial to improve the ability to change direction, 20m and 12m short-distance running. The CG of this study is plyometric training, but in the results, the data of the CG before and after the experiment did not show significant differences in the direction change ability test and the short-distance running test. In particular, there was no significant change in jumping ability after 6 weeks of jump-plyometric training. Can you explain this phenomenon?

Author Response

29-Dec-2022

Dear Editor and dear Reviewer:

We are pleased to resubmit for publication the revised version of Manuscript entitled "Effects of Plyometric Training with Resistance Bands on Neuromuscular Characteristics in Junior Tennis Players". We appreciate the constructive criticisms of the Editor and the reviewers. We have addressed each of their concerns as outlined below.

Responses to the comments from Reviewer 1:

Abstract

CODS should be spelled out in full on the first occurrence.

Done.

"p < .05" requires uniform formatting.

Thank you for this comment. This is uniformly formatted throughout the manuscript.

Introduction

Add research on the use of WearBands™ for muscle training and increase the need for this research.

Thank you for these comments. We have added some research on the use of WearBands as auxiliary training equipment. To the best of our knowledge, no previous research has investigated the effects of plyometric training, using rubber bands with the relevant physical abilities of young tennis players. However, importance of WearBands as an auxiliary tool in training is emphasized in the text. The elastic band presents a new material for plyometric training with load-like weight machines, but it is less expensive and simple to implement. This kind of training can activate all relevant muscles and requires little time. 

Participants

How were the 30 subjects divided into two groups?

Each subject was randomly divided into two groups, CG and TG.

Have you considered the ratio of male to female in the two groups?

We considered the ratio of male to female in each training group.

Measurements

After the intervention experiment, when was the measurement data obtained after the intervention training phase?

The measurement data was obtained right after completing the whole intervention experiment (end of the sixth week).

“There were three of investigators and each of them was charged for the specific group of tests.” Does it mean that the test items are divided into three project groups, and three investigators are responsible for each?

Thank you for pointing this out! Yes, we can say it that way; the test items were divided into three project groups, and three investigators are responsible for each.

Study Design and Procedure

“The 6-week plyometric training is composed of low and moderate levels of plyometric exercises” What are the criteria for distinguishing between low and moderate levels? The training program needs to reflect the low and moderate levels

The criteria for distinguishing between low and moderate levels is the difficulty of the exercise itself. Low level plyometric exercises would include those jumps that can be performed in a high frequency repetition (i.e., jumping rope, ankle / pogo jumps). Moderate level jumps would require a better form of movement, not only in technique but also in mobility and stability, as prerequisites. We did not want to include high level jumps in the program, just because they require an advance mobility and stability and jumping and landing techniques (for example, drop and depth jump) and since our subjects were young athletes, they didn’t have those prerequisites.

“Subjects mostly performed 3 to 4 sets of 4 to 6 exercises with 5 to 10 repetitions with maximum intensity.” How to monitor the implementation of maximum intensity?

Participants were instructed to perform all exercises with maximal effort.

Experimental protocol

The study protocol did not address the use of 3 levels of resistance bands during the intervention. This should be supplemented in the study design. Need to explain the resistance parameters of the following three strength elastic bands?

The elastic bands were of differing elasticity. Green elastic bands (easy resistance), gray elastic bands (low resistance), and yellow elastic bands (moderate resistance) were used for the first and second weeks, the third and fourth weeks, and the fifth and sixth weeks, respectively, to ensure progression. For each color, the training sessions were performed by stretching the band to 75% the first week and 100% for the second week. The elastic band plyometric training began with a resistance of 3.0 kg, with an increase of 1 kg every two weeks to reach a final resistance of 5.0 kg, and the resistance was derived from the manufacturer’s manual, based on the elongation of the band.

Statistical Analysis

 “p ≤ 0.05” Please confirm ≤ or <.

Done.

Results

Table 2. What is the significance of the description of the basic characteristics of the participants? If there are differences in the basic characteristics of the samples, there is no need to continue the analysis.

Completely agree with this comment. There is no significance of the description of the basic characteristics of the participants. Therefore, Table 2. has been deleted.

Table 3. The results section in the manuscript only describes the changes within the group before and after the experiment within the group. Why is there no description of whether there is significance between the groups after the experiment?

In Table 3. it is shown significance interaction within and between factors since 2x2 mixed model ANOVA is considering one repeated measures factor (time – initial and final testing) and one between groups factor (control or experimental group). Regarding this statistical method it already performed between group differences under these interactions. As initially these two groups might differ, it is necessary to observe not only group differences but influence of time (program) on improvement of performance.

Discussion

Many studies mentioned in the introduction have proved that plyometric training is beneficial to improve the ability to change direction, 20m and 12m short-distance running. The CG of this study is plyometric training, but in the results, the data of the CG before and after the experiment did not show significant differences in the direction change ability test and the short-distance running test. In particular, there was no significant change in jumping ability after 6 weeks of jump-plyometric training. Can you explain this phenomenon?

This is a great comment! Well, in CG we found improvement in results after training program in vast majority of variables. However, these improvements were not statistically significant. There are couple of explanation of this phenomenon. Firstly, it is possible that our plyometric program did not significantly emphasize the maximal vertical jumps; secondly, we did not include high level jumps in the program, just because they required an advance mobility and stability and jumping and landing techniques and our young subjects didn’t have those prerequisites, and thirdly, our program was based on a combination of lateral, horizontal, and vertical direction drills, in contrast to previous studies, in which the amount of vertical direction drills was higher.

We thank the editor and the reviewers again for their helpful comments, which we feel have improved our manuscript. We hope that with these modifications, our paper can now be accepted for publication.

Sincerely,

Dario Novak

Reviewer 3 Report

accepted

Author Response

29-Dec-2022

Dear Editor and dear Reviewer:

We are pleased to resubmit for publication the revised version of Manuscript entitled "Effects of Plyometric Training with Resistance Bands on Neuromuscular Characteristics in Junior Tennis Players". We appreciate the constructive criticisms of the Editor and the reviewers. We have addressed each of their concerns as outlined below.

Responses to the comments from Reviewer 3:

Accepted. 

Thank you for all of your valuable comments. We really appreciate your constructive criticisms and we feel your comments have improved our manuscript substantially .

We thank the editor and the reviewers again for their helpful comments, which we feel have improved our manuscript. We hope that with these modifications, our paper can now be accepted for publication.

Sincerely,

Dario Novak
